# Reconstruction of the Long-Term Dynamics of Particulate Concentrations and Solid–Liquid Distribution of Radiocesium in Three Severely Contaminated Water Bodies of the Chernobyl Exclusion Zone Based on Current Depth Distribution in Bottom Sediments

**Alexei Konoplev** [1,*], **Gennady Laptev** [2], **Yasunori Igarashi** [1], **Hrigoryi Derkach** [2], **Valentin Protsak** [2], **Hlib Lisovyi** [2], **Kyrylo Korychenskyi** [2], **Serhii Kirieiev** [3], **Dmitry Samoilov** [3] and **Kenji Nanba** [1]

[1] Institute of Environmental Radioactivity, Fukushima University, Kanayagawa 1, Fukushima 960-1296, Japan; y-igarashi@ipc.fukushima-u.ac.jp (Y.I.); nanba@sss.fukushima-u.ac.jp (K.N.)

[2] Ukrainian Hydrometeorological Institute, Nauki Av., 37, 03028 Kyiv, Ukraine; glaptev@uhmi.org.ua (G.L.); dgrygorii@uhmi.org.ua (H.D.); protsak@uhmi.org.ua (V.P.); hlib@uhmi.org.ua (H.L.); korychenskyi@uhmi.org.ua (K.K.)

[3] Chernobyl Ecocentre, State Agency of Ukraine on Exclusion Zone Management, 07270 Chernobyl, Ukraine; kireev@ecocentre.kiev.ua (S.K.); samoilov@ecocentre.kiev.ua (D.S.)

* Correspondence: r701@ipc.fukushima-u.ac.jp

**Abstract:** Given the importance of understanding long-term dynamics of radionuclides in the environment in general, and major gaps in the knowledge of [137]Cs particulate forms in Chernobyl exclusion zone water bodies, three heavily contaminated water bodies (Lakes Glubokoe, Azbuchin, and Chernobyl NPP Cooling Pond) were studied to reconstruct time changes in particulate concentrations of [137]Cs and its apparent distribution coefficient $K_d$, based on [137]Cs depth distributions in bottom sediments. Bottom sediment cores collected from deep-water sites of the above water bodies were sliced into 2 cm layers to obtain [137]Cs vertical profile. Assuming negligible sediment mixing and allowing for [137]Cs strong binding to sediment, each layer of the core was attributed to a specific year of profile formation. Using this method, temporal trends for particulate [137]Cs concentrations in the studied water bodies were derived for the first time and they were generally consistent with the semiempirical diffusional model. Based on the back-calculated particulate [137]Cs concentrations, and the available long-term monitoring data for dissolved [137]Cs, the dynamics of [137]Cs solid–liquid distribution were reconstructed. Importantly, just a single sediment core collected from a lake or pond many years after a nuclear accident seems to be sufficient to retrieve long-term dynamics of contamination.

**Keywords:** Chernobyl NPP; radiocesium; dissolved; particulate; lakes; cooling pond

## 1. Introduction

Thirty-five years after the Chernobyl nuclear power plant (ChNPP) accident, studies of radioactive contamination of water bodies continue to be of importance and relevance due to the need to understand long-term processes and dynamics. Closed and semi-closed lakes and ponds were found to be most sensitive to radioactive contamination, as evidenced by numerous studies conducted in the Chernobyl contaminated areas [1–5] and the Fukushima Dai-ichi nuclear power plant zone [6–10], as well as in the PA Mayak area [11–13] and in the USA including Savannah River site [14–16]. Despite a vast number of studies of radionuclide behavior in such water bodies, most of them lacked broad temporal coverage and did not deal with long-term changes in radionuclide concentrations. This is especially true for particulate concentrations in the Chernobyl zone water bodies, with the monitoring system primarily focused on dissolved concentrations as being the most important

in terms of radionuclide mobility and bioavailability [17,18]. In our work, particulate concentration as a function of time is reconstructed from radionuclide depth distribution in bottom sediments, assuming certain conditions are satisfied for sediment column formation. Such thinking was previously used for Chernobyl-derived $^{137}$Cs profiles in the bottom sediments of Shchekino dam reservoir on Upa River [19,20] and for Fukushima-derived $^{137}$Cs in bottom sediments of Ogaki dam on Ukedo River [21] to estimate radionuclide concentrations in rivers. This study, to our knowledge, is the first attempt to apply the approach for lakes and artificial pond.

Immediately after the accident, the primary pathway of radioactive contamination for water bodies is atmospheric fallout directly on the water surface [5,22]. The main long-term source of sediments and sediment-associated radionuclides in lakes is surface runoff from the contaminated catchment [5,23–25], with suspended particles delivered from the topsoil layer on the catchment. Numerous studies of wash-off of different origin contaminants have shown that the effective soil layer depth interacting with surface runoff is only the top several millimeters (up to 1 cm) [26–31]. The radionuclide concentration in the "contact" topsoil layer decreases with time due to radioactive decay and vertical migration to deeper layers [32–36]. As a consequence, the particulate concentration of radionuclides in lakes and ponds decreases, and, accordingly, the dissolved concentration also decreases. Suspended particles settle continuously on the bottom, and thereby the depth profile of sediment-associated $^{137}$Cs is formed in bottom sediments. The highest concentrations of radionuclide in the sediment profile can be attributed to the initial period or the first year after the accident. This holds true, however, provided mixing of particles in the bottom sediment column is negligible and the radionuclide is strongly bound by sediments.

The ability of suspended sediments to bind a radionuclide is usually characterized by the apparent distribution coefficient $K_d$, which, by definition, is the ratio of radionuclide particulate concentration $c_p$ to its dissolved concentration $c_d$ at equilibrium or steady-state condition [37–39]:

$$K_d = \frac{c_p}{c_d} \qquad (1)$$

The requirement of radionuclide strong binding to sediments is fulfilled when $K_d > 10^4$ L/kg [19–21].

When mixing of sediments in the bottom depth profile is insignificant and radionuclides are strongly bound to sediments, the maximum radionuclide concentration in its depth distribution should be well marked with a sharp front below. Reasoning from this, many authors conducted dating of bottom sediments and determined average sedimentation rates in lakes [40–43]. Determination of the sedimentation rate based on the position of the radionuclide concentration peak in the sediment profile is common practice. Such studies, however, are focused on sediments and their dating exclusively and do not relate the radionuclide depth profile in bottom sediments to temporal changes in radioactive contamination of the water body. In our work, we use $^{137}$Cs depth distribution in bottom sediments to reconstruct activity concentrations in water as a function of time in lakes and ponds, which have not been attempted before.

The sediment profile formed since the accident shows how the particulate concentration of $^{137}$Cs in the water body changed with time, given the highest $^{137}$Cs concentration in the depth profile corresponds to its particulate concentration in the initial period after the accident, and $^{137}$Cs concentration in the top sediment layer represents the current particulate $^{137}$Cs concentration in the water column. Changes in the particulate $^{137}$Cs concentration with time can be reconstructed from the profile above the peak, given data on annual average sedimentation rate are available or, as a first approximation, assuming this parameter to be constant over the time after the accident [19,21].

Long-term dynamics of $^{137}$Cs concentrations in water bodies can be described using the earlier proposed semi-empirical diffusional model [44,45]. According to the model, for water bodies, where a major source of suspended material is the top layer of catchment soil,

the long-term dynamics of $^{137}$Cs particulate concentration $c_p$ and dissolved concentration $c_d$ can be described by the following equations:

$$c_p(t) = \frac{\sigma}{\rho\sqrt{\pi D_{eff} t}} e^{-\lambda t} = c_p^0 \frac{e^{-\lambda t}}{\sqrt{t}}; \qquad c_p^0 = \frac{\sigma}{\rho\sqrt{\pi D_{eff}}} \tag{2}$$

$$c_d(t) = \frac{\sigma}{\rho K_d\sqrt{\pi D_{eff} t}} e^{-\lambda t} = c_d^0 \frac{e^{-\lambda t}}{\sqrt{t}}; \quad c_d^0 = \frac{\sigma}{\rho K_d\sqrt{\pi D_{eff}}} \tag{3}$$

where $\sigma$ is the catchment averaged $^{137}$Cs deposition (Bq/m$^2$); $\rho$ is the catchment averaged bulk density of the topsoil layer (g/m$^3$); $D_{eff}$ is the catchment averaged effective coefficient of $^{137}$Cs dispersion in the topsoil layer (m$^2$/y); $\lambda$ is the rate constant of $^{137}$Cs radioactive decay equal to 0.023 y$^{-1}$; $K_d$ is the $^{137}$Cs apparent distribution coefficient in the suspended sediment–water system in the water body (m$^3$/g) and $t$ is the time since the Chernobyl accident (y).

This model enables predicting particulate and dissolved $^{137}$Cs concentrations in lakes and ponds for the mid- and long-term phases after a nuclear accident, using, besides $^{137}$Cs deposition, two key physicochemical parameters of $^{137}$Cs dispersion and distribution in the sediment–water system. The particulate $^{137}$Cs concentration is determined by the effective dispersion coefficient $D_{eff}$ in the catchment topsoil, depending on $^{137}$Cs speciation, sorption, and fixation ability of catchment soil, and climatic conditions (e.g., mean annual rainfall, mean annual air temperature, etc.) [31–36], whereas the dissolved $^{137}$Cs concentration, along with $D_{eff}$, is controlled by the distribution coefficient $K_d$ in the sediment–water system [3,45].

The present study aims (1) to reconstruct $^{137}$Cs particulate concentrations in severely contaminated water bodies in the Chernobyl exclusion zone (ChEZ), using the $^{137}$Cs depth distributions in the bottom sediment cores collected; (2) to estimate time dependence of $^{137}$Cs apparent distribution coefficient in these water bodies; and (3) to test the applicability of semi-empirical diffusional model to describe long-term dynamics of particulate and dissolved $^{137}$Cs in heavily contaminated water bodies.

## 2. Materials and Methods

### 2.1. Study Sites

For study purposes, the three most heavily contaminated water bodies in the ChNPP exclusion zone were selected: Typical eutrophic oxbow lakes of the Pripyat River floodplain, namely, Glubokoe (left bank of the Pripyat River) and Azbuchin (right bank of the Pripyat River), and a man-made reservoir, the Chernobyl NPP cooling pond (Figure 1). Table 1 includes the main characteristics of the studied water bodies.

The soils in the studied catchments are mainly alluvial-soddy-meadow and soddy-weakly podzolic soils with a sandy easily permeable bedrock. Part of the Glubokoe catchment is occupied by man-made plantations of deciduous trees. The depth of groundwater occurrence is 1.5–2.0 m depending on the season since the lakes are hydraulically connected to the Pripyat River channel. Bottom sediments are made of black sandy mud, and a layer of dense sandy-gley parental C-subsoil occurs at the core bottom.

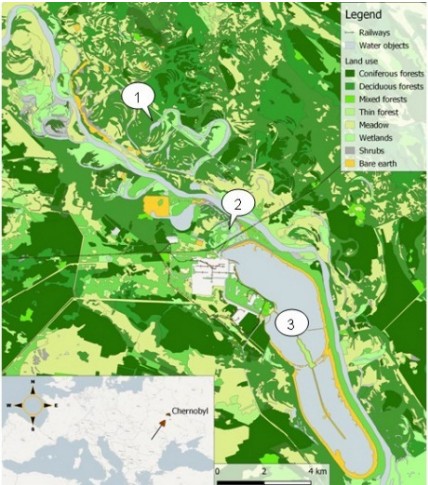

**Figure 1.** Location of the studied water bodies in the Chernobyl exclusion zone on the map of land use: 1—Lake Glubokoe; 2—Lake Azbuchin; 3—Chernobyl NPP cooling pond.

**Table 1.** Characteristics of the studied water bodies in the Chernobyl exclusion zone.

| Water Body | Lake Glubokoe | Lake Azbuchin | Cooling Pond |
|---|---|---|---|
| Coordinates | 51.444° N 30.065° E | 51.406° N 30.117° E | 51.354° N 30.166° E |
| $^{137}Cs$ mean deposition, $MBq/m^2$ | 11 ± 5 [a] | 7.6 ± 3.2 [a] | 4.2 ± 1.3 [b] |
| Water surface area, $km^2$ | 0.17 | 0.23 | 22.6 |
| Maximum depth, m | 7.1 | 5.5 | 17.0 |
| Average depth, m | 3.0 | 2.5 | 6.2 |
| Reference | [46] | [47] | [18] |

[a] Data of this work for 2018; [b] Middle part (dyke).

## 2.2. Sampling

The soil cores from Lakes Glubokoe and Azbuchin catchments 68 to 102 cm long were collected to determine $^{137}Cs$ inventories using a DIK-110C liner soil sampler (DAIKI, Japan) with a plastic cylinder insert of 5 cm in diameter. The soil cores were sliced into 2 cm layers. Bottom sediment cores were collected in August 2018 using a gravity sampler designed by Ukrainian Hydrometeorological Institute (UHMI). Sampling was conducted at the deepmost places where sediments could potentially accumulate and where horizontal water flows are minimal. Depth was determined by echo sounding of the bottom using FishElite 480 (Eagle, USA). Water temperature and dissolved oxygen (DO) in water bodies were measured using submersible multitester U-52 (HORIBA, Japan). In the cooling pond (CP), the location of bottom sediment core sampling was tied to a small old floodplain no-name lake, and its coordinates were taken from a military topographic map from 1926, well before the construction of the CP.

## 2.3. Sample Analysis

The water samples were filtered in situ through sandwich-type Petryanov filters (with a pore size of 0.5 μm) to separate suspended material from solution [48]. Dissolved $^{137}Cs$ was immobilized by two sequential cartridges of cellulose-based and iron hexacyanoferrate-impregnated ANFEZH sorbent (EKSORB Ltd., Ekaterinburg, Russia) [10,49,50]. Alternatively, the filtrate was collected into plastic containers and 1 mL of 6M nitric acid was added for direct determination of activity. Gamma spectrometry with GMX-40-LB (Ortec, USA) high purity germanium detector (HPGe) was used to determine $^{137}Cs$ activity concentration in soil and bottom sediment samples, filtrate, and suspended material, as well as $^{241}Am$ in the soil and bottom sediment samples [51]. Detector calibration was implemented using NIST-, IAEA-, and NPL UK-certified reference materials. Corrections

were made for the effect of self-absorption of low energy g-rays within the sample matrix using mass-attenuation parameters determined by the method reported in [52], considering actual composition of the sediment matrix ($CaCO_3$, mineral and organic matter). All measurements were performed in the analytical laboratories of the UHMI (Kyiv) and Eco-centre (Chernobyl). The reported uncertainty of analytical results, presented as expanded uncertainty calculated from combined standard uncertainty using a coverage factor k = 2 at level of confidence of 95%, did not exceed ±10%. Most contribution to combined standard uncertainty comes from spectral peak statistics, calibration curve fitting, and reported uncertainty of calibration standards.

## 3. Results

The characteristics of bottom sediment cores collected from the three water bodies for successive analysis are provided in Table 2. Depth distributions for [137]Cs and [241]Am in bottom sediments in the accumulation zones are shown in Figure 2. [137]Cs and [241]Am in the investigated water bodies are strongly bound by solid soil and sediment particles, and the values of their solid–liquid distribution coefficient in suspended matter–water system $K_d > 10^4$ L/kg [37].

**Table 2.** Characterization of the bottom sediment cores.

| Water Body | Lake Glubokoe | Lake Azbuchin | Cooling Pond |
|---|---|---|---|
| Sampling date | 25.07.2018 | 24.07.2018 | 05.06.2018 |
| Sampling location coordinates | N51.34583 E30.16242 | N51.4057 E30.1119 | N51.3458 E30.1624 |
| Depth at sampling location, m | 6.3 | 4.2 | 8.6 |
| Core length, cm | 41 | 62 | 68 |
| Mean sedimentation rate [a], $kg/m^2y$ (cm/y) | 1.65 (1.1) | 1.2 (1.4) | 3.2 (1.7) |
| [137]Cs inventory [b], $MBq/m^2$ | 18 ± 3.6 | 17 ± 3.4 | 63 ± 12 |
| [241]Am inventory [b], $MBq/m^2$ | 0.8 ± 0.16 | 0.5 ± 0.1 | 1.7 ± 0.35 |

[a] Determined by position of the [137]Cs peak attributed to the first year after the accident 1986–1987; [b] Expanded uncertainty at coverage factor k = 2 and level of confidence of 95%.

As shown in Figure 2, the [137]Cs and [241]Am profiles have a pronounced sharp maximum from below. Presumably, the activity peak for these radionuclides was formed in the first year after the accident when radioactive particles fell out and settled on the bottom. Right below the peak, the [137]Cs and [241]Am activity concentrations in the profile are about an order of magnitude lower, which suggests that the suspended material deposited on the bottom did not mix with the underlying layers of the bottom sediments, and the dispersion of radionuclides through interstitial water in the sediments is rather insignificant. Lack of mixing in the deep-water zones of the lakes was also supported by numerous measurements, showing the deficit of DO there, to the point of its total absence. In the absence of oxygen, vital activity of biota is minimized and as a result bioturbation, as one of bottom sediment mixing drivers, does not occur. We admit that horizontal redistribution of sediments is possible, to some extent, along the bottom surface after their sedimentation. This, however, will not change the general vertical profile of radionuclides formed over many years.

It should be noted that in the following discussion we use only data on [137]Cs distribution in bottom sediments. Results of [241]Am measurements and their comparison with [137]Cs were provided exclusively to illustrate that the vertical distribution of [137]Cs in bottom sediments is determined by settling of carrier particles. Absolute values of activity concentrations for these two radionuclides differ for more than one order of magnitude. Nevertheless, the profiles of [137]Cs and [241]Am in relative units appeared to be very similar. Both radionuclides are strongly bound by sediments for solute transport to play negligible role in their vertical distribution in bottom sediments, though [241]Am is even more strongly bound by sediments than [137]Cs [37]. In this context, vertical distributions of [137]Cs in bot-

tom sediment cores collected in 2018 may represent the time dependence of $^{137}$Cs activity concentrations in suspended sediments deposited on the bottom after the accident.

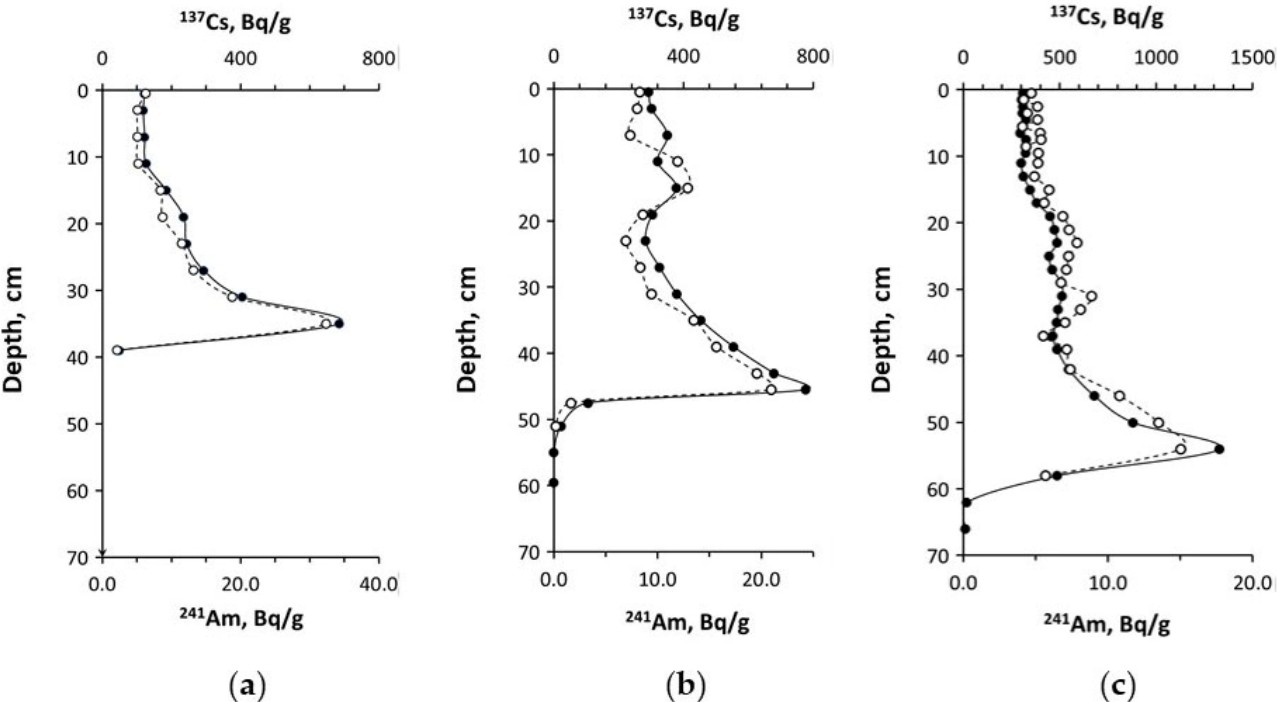

**Figure 2.** $^{137}$Cs (●) and $^{241}$Am (○) depth distributions in the bottom sediment cores collected in the studied water bodies: (**a**)—Lake Glubokoe, (**b**)—Lake Azbuchin, (**c**)—Chernobyl NPP cooling pond.

## 4. Discussion

### 4.1. Reconstruction of Long-Term Dynamics of Particulate $^{137}$Cs Activity Concentrations in Studied Water Bodies

The $^{137}$Cs activity concentration in the sediment core layers above the peak slowly declines up to the surface sediment layer, as can be seen from Figure 2, which reflects temporal changes in particulate concentrations in the water column after the accident. Some minor deviations can be noticed for Lake Azbuchin and CP, yet not contradicting the general trend. There are also secondary maxima in the $^{137}$Cs activity concentrations, which are likely to be associated with water level manipulation and other remediation activities, as well as changes in the ChNPP operational mode [18].

The obtained $^{137}$Cs core profiles provided a basis for reconstructing long-term dependence of its particulate activity concentration in the water column. This is especially important since throughout the years after the accident, this crucial characteristic of radioactive contamination was not monitored on a regular basis but measured only occasionally in selected water bodies within some projects [18,44,45,53].

For reconstruction, the whole $^{137}$Cs profile from the peak to the sediment surface is split into layers, representing sediments deposited on the bottom over a respective time interval. It can be assumed that the sediments accumulate evenly in a timewise manner, i.e., about the same amount of sediments depositing on the bottom each year. Even though this is an approximation, ample corroborating data can be found in the literature. For example, studies on bottom sediments dating in Brno dam reservoir (Czech Republic) for over almost 70 years (from 1939 to 2007) showed that the sedimentation rate there was quite stable, varying in a narrow range from 3.1 to 3.4 cm/y [43].

Given the differences in bottom sediment density with depth, it seems reasonable to use the dependence of $^{137}$Cs activity concentration on mass depth in kg/m$^2$ rather than depth in cm. The mean annual sedimentation rates for the studied water bodies both in

kg/m²y and in cm/y, calculated by the position of the $^{137}$Cs peak attributed to the first year after the accident (1986–1987), are presented in Table 2.

When reconstructing the time dependence of $^{137}$Cs activity in suspended sediments, we corrected for $^{137}$Cs radioactive decay over the time elapsed since the sediment deposition on the bottom. The $^{137}$Cs activity in the profile peak decreased by approximately a factor of two over 32 years since sediment deposition, given the half-life of $^{137}$Cs is 30.2 years. For decay correction, we multiplied the measured activity in the sediment layer, as of the sampling date in 2018 by $e^{\lambda(2018-t)}$, where $t$ is the reconstructed year for the given sediment layer, and $\lambda = 0.023$ y$^{-1}$ is the $^{137}$Cs radioactive decay rate constant. The decay corrected $^{137}$Cs profiles for the bottom sediments in the studied water bodies are shown in Figure 3 versus uncorrected profiles (i.e., the depth profiles of $^{137}$Cs activity concentrations as of the core sampling date).

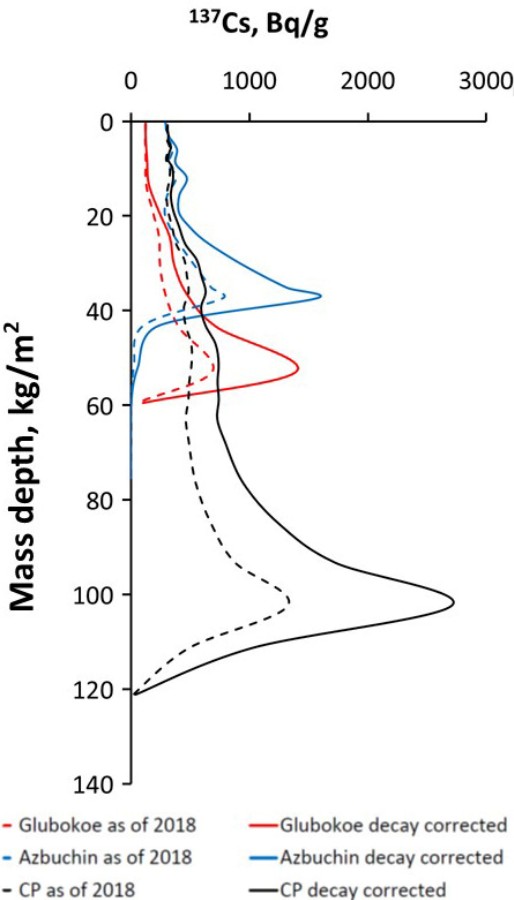

**Figure 3.** $^{137}$Cs activity concentration profiles corrected for radioactive decay over the time since deposition as a function of mass depth of bottom sediments versus uncorrected profiles, as of the sampling date in 2018.

The reconstructed time dependences of $^{137}$Cs particulate activity concentration in the three studied water bodies since the accident are shown in Figure 4. The significance of this data is that it seems to be the first and, so far, the only estimate of long-term dependence of $^{137}$Cs particulate concentration in the heavily contaminated water bodies of the ChNPP exclusion zone. Knowing $^{137}$Cs particulate concentrations is necessary not only for a better understanding of the radionuclide behavior in the soil–water system, but also for assessment of radionuclide transfer to biota and food chain migration.

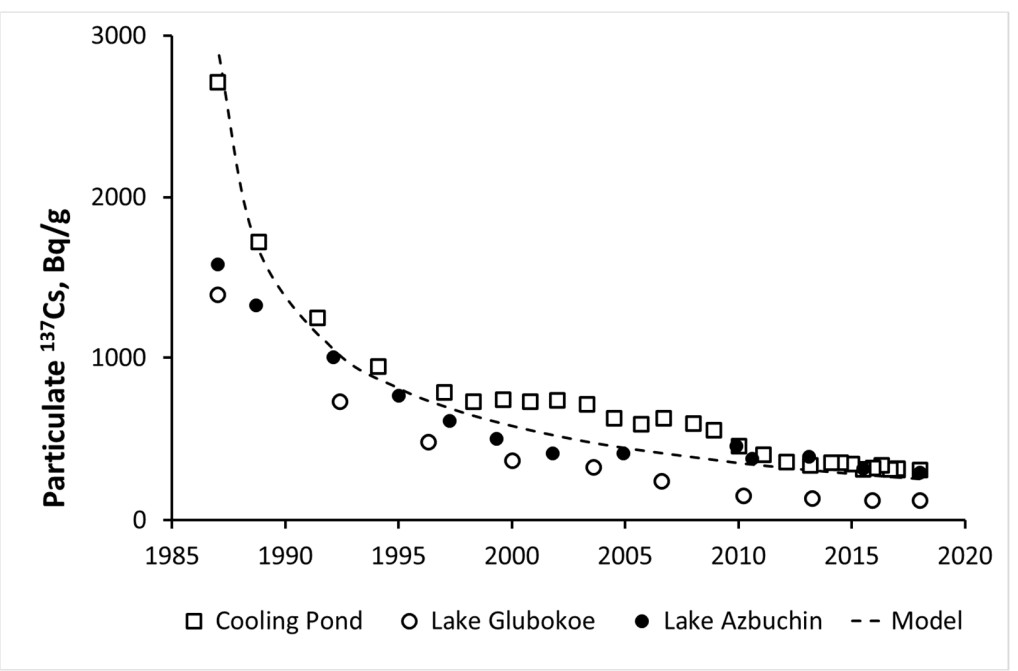

**Figure 4.** Time dependence of particulate $^{137}$Cs activity concentrations in Lakes Glubokoe, Lake Azbuchin, and Cooling Pond after the accident in 1986, back calculated from the $^{137}$Cs depth distribution in the bottom sediments collected in 2018 in deep-water zones. Dashed line—calculation by the semiempirical diffusional model (Equation (2)) at $c_p^0 = 3000$ Bq·(y)$^{1/2}$g$^{-1}$.

The reconstructed data, in addition, can be used for testing the semi-empirical diffusional model (Equation (2)) accounting for dynamics of particulate $^{137}$Cs concentrations in water bodies [44,45]. The obtained data help to ascertain that the processes and mechanisms described by this model really occur in the studied water bodies and therefore the model can be used for prediction purposes. The $^{137}$Cs particulate concentrations as a function of time in the water bodies, calculated by the model at $c_p^0 = 3000$ Bq·(y)$^{1/2}$g$^{-1}$, are shown in Figure 4 (dashed line).

In the calculations of $c_p^0$, we used the $^{137}$Cs effective dispersion coefficient in soil 0.1 cm$^2$/y, which corresponds to the lower bound of the range of $D_{eff}$ values in the ChEZ soils [33,36,54]. The low value of $D_{eff}$ was taken since $^{137}$Cs deposited on these catchments was incorporated in persistent fuel particles characterized by slow disintegration and low mobility [18,55,56].

### 4.2. Reconstruction of Long-Term Dynamics of Apparent $^{137}$Cs Distribution Coefficient $K_d$ ($^{137}$Cs) in Studied Water Bodies

Based on the reconstructed $^{137}$Cs particulate concentrations and dissolved $^{137}$Cs concentrations obtained by monitoring [17,18] we estimated $K_d$ for the time period from 1986 to 2018. The results for the three water bodies are shown in Figure 5. Earlier studies of the Chernobyl rivers revealed no time trend in the annual average $K_d$ ($^{137}$Cs) in the suspended matter–water system after the accident [44,45]. As shown in Figure 5, this is true for Lakes Glubokoe and Azbuchin, for which the multi-year average values for $K_d$ ($^{137}$Cs) are $(3.5 \pm 0.7) \times 10^4$ L/kg and $(5.5 \pm 2.1) \times 10^4$ L/kg, respectively. However, the picture is different for the CP, in which the apparent $K_d$ ($^{137}$Cs) increased from $5.5 \times 10^4$ L/kg in 1986–87 to $3 \times 10^5$ L/kg in 1997, and reached a plateau afterwards.

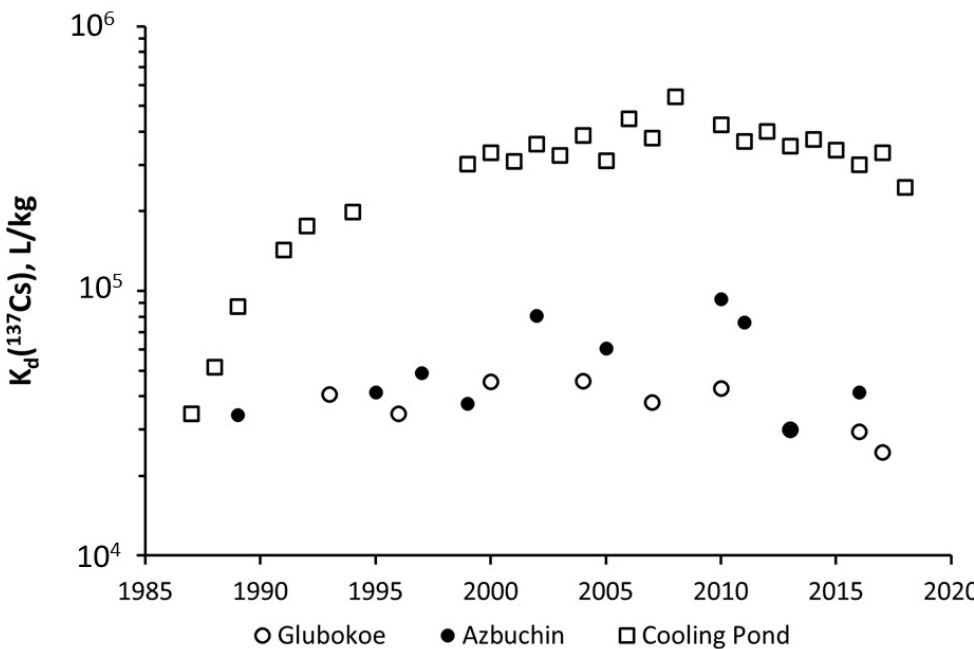

**Figure 5.** Time changes in the $^{137}$Cs apparent distribution coefficient $K_d$ ($^{137}$Cs) estimated from reconstructed particulate $^{137}$Cs concentrations and data of long-term monitoring of dissolved $^{137}$Cs in the studied water bodies.

The obtained $K_d$ ($^{137}$Cs) time dependence for the cooling pond is most likely to be due to CP maintenance features. The mode of ChNPP operations changed significantly and consequently the circulation, thermal, hydrological, and hydro-chemical conditions in CP changed too. In the years following the initial phase, the $K_d$ ($^{137}$Cs) value in the suspended matter–water system of CP was approximately an order of magnitude higher than those typical of other water bodies in the ChNPP zone [18].

With $K_d$ ($^{137}$Cs) time dependence reconstructed, it became possible to calculate temporal changes in dissolved $^{137}$Cs in the studied water bodies using the semi-empirical diffusional model (Equation (3)). Figure 6 shows available monitoring data on dissolved $^{137}$Cs activity concentrations over time in the studied water bodies [17,18] versus calculations by the diffusional model. In model calculations of the dissolved $^{137}$Cs concentrations, $K_d$ ($^{137}$Cs) was taken to be constant and equal to $3.5 \times 10^4$ L/kg for Lake Glubokoe and $5.5 \times 10^4$ L/kg for Lake Azbuchin. For the CP, long-term monitoring data show an abnormally quick reduction in the dissolved $^{137}$Cs concentration from 1986 to 1997 [18], which is possibly associated with a sharp increase in $^{137}$Cs apparent distribution coefficient, as supported by Figure 5. When modeling the dissolved $^{137}$Cs concentration time dependence, we therefore assume $K_d$ ($^{137}$Cs) in CP to grow from $5.5 \times 10^4$ L/kg (the value typical for Lake Azbuchin located nearby and groundwater connected with CP) in 1986–1987 to $3.0 \times 10^5$ L/kg in 1997, remaining constant since then. The results of our calculations of dissolved $^{137}$Cs activity concentrations time dependences by the diffusional model are consistent with the available monitoring data (Figure 6).

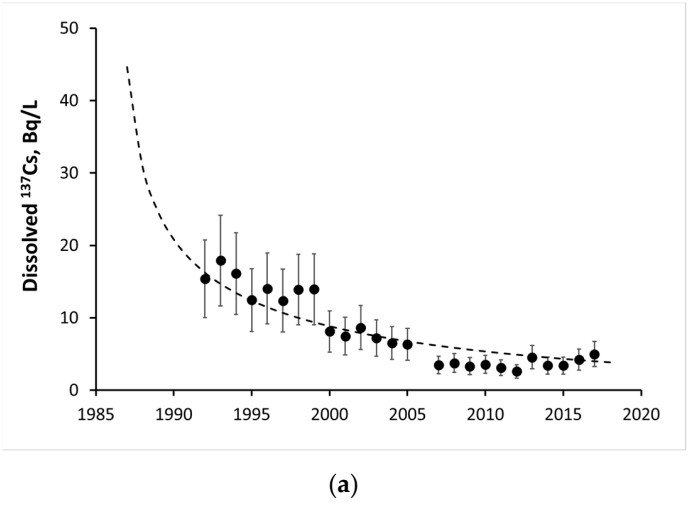

(**a**)

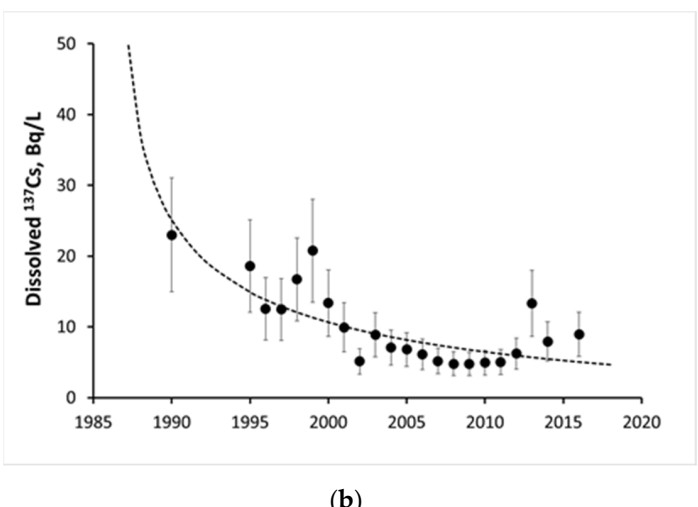

(**b**)

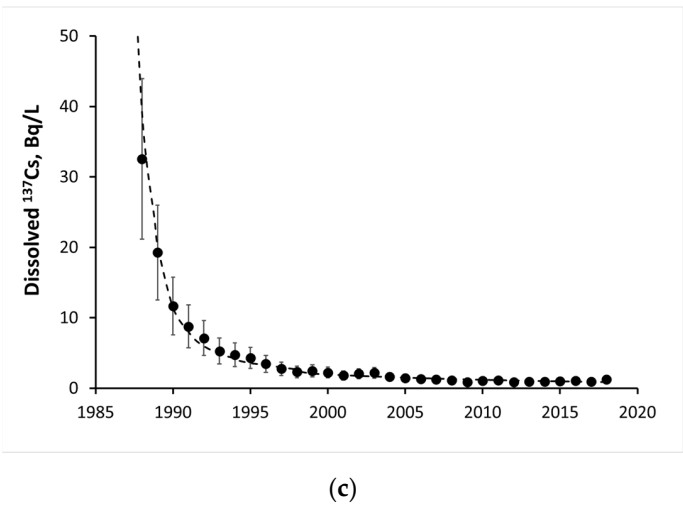

(**c**)

**Figure 6.** Time dependence of annual averaged monitoring data on dissolved $^{137}$Cs activity concentrations in the studied water bodies (black circles) versus calculations by the semiempirical diffusional model (dashed lines): (**a**)—Lake Glubokoe, (**b**)—Lake Azbuchin, and (**c**)—ChNPP cooling pond. In model calculations $K_d$ ($^{137}$Cs) was taken to be constant and equal to $3.5 \times 10^4$ L/kg for Lake Glubokoe and $5.5 \times 10^4$ L/kg for Lake Azbuchin. $K_d$ ($^{137}$Cs) in the cooling pond was assumed to increase from $5.5 \times 10^4$ L/kg in 1986–1987 to $3.0 \times 10^5$ L/kg in 1997, remaining constant since then. $D_{eff}$ was taken to be 0.1 cm$^2$/y for all three water bodies.

We admit that certain assumptions were used in our reconstruction of long-term dynamics of [137]Cs concentrations, leading to uncertainty in results. Specifically, it is required that sediments settling on the bottom should not mix in the course of sediment profile formation throughout the time, and the other condition is that the radionuclide be strongly bound to sediments. In using the proposed approach, it is essential to make sure that these conditions are satisfied. Even though such requirements are constraints of the method, evidence from experience suggests that it is also so in reality. It is also worth pointing out that our study is concerned with temporal trends of sedimentation based on using mean annual values, which results in smoothing over variations in sedimentation rate within a specific year. Actually, the effect of variation smoothing was demonstrated for a number of lakes across the world with relatively non-uniform distribution of precipitation within a year [40,43,57]. We supposed that the same would be true for ChEZ water bodies under study and the obtained reconstruction results were not contrary to what we expected.

Strictly speaking, the accuracy of the proposed reconstruction method can be improved in the future by accounting for annual precipitation variations from year to year. For the ChEZ, however, annual precipitation generally differs from the mean annual precipitation by not more than 15–20%, and therefore the assumption in our reconstruction that the sedimentation rate does not change much from year to year seems very reasonable. Parameterization of mean annual sedimentation rate through rainfall erosivity factor R, based on Universal Soil Loss Equation (USLE) [58], can help to obtain an even more realistic sedimentation history. Use of such a parameterization of sedimentation rate time dependence would reduce uncertainty in reconstruction results. At this stage it is important for us to set out the concept underlying our method, and in future it can be further developed and improved.

The proposed method, when used jointly with the diffusional model, enables not only reconstructing dynamics of contamination in the past, but also predicting future trends in contamination.

## 5. Conclusions

The [137]Cs depth distribution in bottom sediment cores collected more than 30 years after the Chernobyl accident in deep parts of heavily contaminated Lakes Glubokoe and Azbuchin and the cooling pond was used to reconstruct particulate [137]Cs concentrations as a function of time. To our knowledge, the obtained estimates of particulate [137]Cs concentrations and their temporal trends in these heavily contaminated water bodies are the first of its kind for the Chernobyl exclusion zone. Knowing [137]Cs particulate concentrations is necessary not only for better understanding and prediction of the radionuclide behavior in the soil–water system, but also for assessment of radionuclide transfer to biota and food chain migration.

Reconstruction of temporal changes in particulate [137]Cs concentration was based on the understanding that each bottom sediment layer can be taken as representing suspended sediments deposited on the bottom during a certain time interval after the accident. Just a single sediment core collected in the deep–water accumulation zone of lake or pond many years after a nuclear accident appears to be sufficient to retrieve long-term dynamics of contamination.

The apparent distribution coefficient $K_d$ ([137]Cs) in the suspended matter–water system, derived using reconstructed [137]Cs particulate concentrations and available monitoring data for dissolved [137]Cs, was found to remain quite constant with time for Lakes Glubokoe and Azbuchin. On the other hand, $K_d$ ([137]Cs) in the cooling pond increased from $5.5 \times 10^4$ to $3.0 \times 10^5$ L·kg$^{-1}$ during the first 10 years after the accident, but afterwards remained constant until very recently. This means that in the long-term [137]Cs mobility and bioavailability in Lakes Glubokoe and Azbuchin can be expected to be much higher than in the cooling pond.

The proposed method enables, based on cesium distribution profile in bottom sediments, reconstructing temporal changes in water-body radioactive contamination for the

years following the accident, which is particularly important when regular monitoring data are not available. We admit that the proposed method involves certain assumptions leading to uncertainty in results. It is required that, as the sediment column is formed, sediments settling on the bottom in different time intervals should not mix, and the other condition is that the radionuclide is strongly bound to sediments. These assumptions, however, seem to be quite realistic. Even though these requirements are constraints on the method, it is important that they are satisfied in performing analysis and calculations.

**Author Contributions:** Conceptualization, A.K. and G.L.; methodology, A.K. and G.L.; software, H.L.; validation, A.K., G.L. and V.P.; formal analysis, A.K.; investigation, H.D., K.K., Y.I. and D.S.; resources, K.N. and S.K.; data curation, A.K. and G.L.; writing—original draft preparation, A.K.; writing—review and editing, G.L. and Y.I.; visualization, H.L.; supervision, K.N. and S.K.; project administration, Y.I.; funding acquisition, K.N. and A.K. All authors have read and agreed to the published version of the manuscript.

**Funding:** This research was partially funded by the Science and Technology Research Partnership for Sustainable Development, the Japan Science and Technology Agency/Japan International Cooperation Agency (SATREPS), grant number JPMJSA1603; the Japan Society for the Promotion of Science, Grant-in-aid for Scientific Research (KAKENHI B), grant number 18H03389; and by the Environment Research and Technology Development Fund of the Environmental Restoration and Conservation Agency of Japan, grant number JPMEERF20211R03.

**Institutional Review Board Statement:** Not applicable.

**Informed Consent Statement:** Not applicable.

**Data Availability Statement:** The data presented in this study are available on request from the corresponding author.

**Acknowledgments:** The authors are grateful to Dyvak T.I. and Derkach A.N. from UHMI for the analytical support.

**Conflicts of Interest:** The authors declare no conflict of interest. The funders had no role in the design of the study; in the collection, analyses, or interpretation of data; in the writing of the manuscript, or in the decision to publish the results.

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
