# Peer review of "Reconstruction of the Long-Term Dynamics of Particulate Concentrations and Solid–Liquid Distribution of Radiocesium in Three Severely Contaminated Water Bodies of the Chernobyl Exclusion Zone Based on Current Depth Distribution in Bottom Sediments"

_land, doi:10.3390/land11010029_

Round 1

Reviewer 1 Report

  1. Please explain combined standard uncertainty at confidence probability in detail. What is the purpose authors like to present?
  2. To give a clear insight that the study reconstruct the long-term dynamics pf particulate Cs-137 activity concentrations and Cs-137 Kd value in studied water bodies.
  3. It is unclear, what reader can follow the study in the future. I cannot find significant contribution of this study.

Author Response

We thank the reviewer for the time spent on the review and reading of our text. We appreciate the comments and suggestions which have been helpful in improving the manuscript.

Please see below a combined point-by-point response to address all reviewers’ comments.

Reviewers’ comments are shown in blue Italic.

  1. Please explain combined standard uncertainty at confidence probability in detail. What is the purpose authors like to present?

Response (1)

Uncertainty is a crucial indicator of the reliability of measurements that our study relies on. The uncertainty associated with all our analytical results was expanded uncertainty (as defined in the ISO 1993) which is calculated from combined standard uncertainty using the coverage factor of 2, giving a level of confidence of 95%. The main contributors to combined standard uncertainty are peak statistics, calibration curve fitting and uncertainty of calibration sources (QUAM:2000.1).

  1. To give a clear insight that the study reconstruct the long-term dynamics pf particulate Cs-137 activity concentrations and Cs-137 Kd value in studied water bodies.

Response (2)

We revised the text (abstract and Introduction) to emphasize that the central objective of the paper is developing a method for reconstruction of particulate Cs-137 and its apparent distribution coefficient Kd as a function of time in three severely contaminated water bodies of ChEZ on the basis of current Cs-137 vertical distribution in bottom sediments. Particulate Cs-137 activity concentrations in water bodies were not monitored systematically and therefore systematic data on Kd and its long-term dynamics were not available either. However, data on dynamics of particulate Cs-137 and Kd are essential for understanding basic mechanisms of radiocesium behavior in the soil-water environment and dose assessment for fish and other biota inhabiting water bodies.

  1. It is unclear, what reader can follow the study in the future. I cannot find significant contribution of this study.

Response (3)

This is the first reconstruction of temporal changes in particulate 137Cs concentration in water bodies based on the vertical distribution in bottom sediments. What is also important is that the method can be used in the future in case monitoring data on radionuclide concentrations are not available. Jointly with the semi-empirical model, the approach enables the prediction of contamination in the future.

Once again, we appreciate the reviewer’s comments and recommendations.

Reviewer 2 Report

The manuscript entitled “Reconstruction of the long-term dynamics of particulate concentrations and solid-liquid distribution of radiocesium in water bodies of the Chernobyl exclusion zone based on its current depth distribution in bottom sediments” presents the research related to the radiocesium pollution in water bodies of exclusion zone in Chernobyl.  The topic is in line with the tendency of research conducted in the zone after nucleat a nuclear disaster, aimed to describe the distribution of radiocesium on the depth in bottom sediments.  Temporal trends of particulate 137Cs concentrations were derived for the water bodies for the first time, and they were found to be consistent with the semiempirical diffusional model. Based on the back-calculated particulate 137Cs concentrations and available long-term monitoring data for dissolved 137Cs, the dynamics of 137Cs solid-liquid distribution was reconstructed. Importantly, the procedure gave possibility to retrieve long-term dynamics of contamination on the basis of a single sediment core collected from a lake or pond many years after a nuclear disaster.  

General comments:

  • In my opinion the Abstract could be revised and modified. 
  • The literature review seems to be very limited. I suggest authors to elaborate more on the topic and refer to many more scientific research related to the paper subject.
  • The literature review should be based on the most recent scientific data. Now, more than 50% of cited references are old. Please improve that.
  • It is not clear what is the hypothesis of performed study.
  • The authors should enhance the discussion and highlight the importance of performed study in terms of the sustainability.
  • The novelty and limitations of the performed research should be delineated.
  • The title indicates that the study was be comprehensive, but after reading the text, the reader may have impression that is still lack of mentioned comprehensiveness, especially in regards to the literature review and the discussion on the outcomes.

Author Response

We thank the reviewer for the time spent on the review and careful reading of our text. We appreciate the insightful, constructive comments and suggestions which have been very helpful in improving the manuscript.

Please see below a combined point-by-point response to address all reviewers’ comments. Reviewers’ comments are shown in blue Italic.

The manuscript entitled “Reconstruction of the long-term dynamics of particulate concentrations and solid-liquid distribution of radiocesium in water bodies of the Chernobyl exclusion zone based on its current depth distribution in bottom sediments” presents the research related to the radiocesium pollution in water bodies of exclusion zone in Chernobyl. The topic is in line with the tendency of research conducted in the zone after a nuclear disaster, aimed to describe the distribution of radiocesium on the depth in bottom sediments. Temporal trends of particulate 137Cs concentrations were derived for the water bodies for the first time, and they were found to be consistent with the semiempirical diffusional model. Based on the back-calculated particulate 137Cs concentrations and available long-term monitoring data for dissolved 137Cs, the dynamics of 137Cs solid-liquid distribution was reconstructed. Importantly, the procedure gave possibility to retrieve long-term dynamics of contamination on the basis of a single sediment core collected from a lake or pond many years after a nuclear disaster.

Response (1)

We thank the reviewer for the understanding of the essence of our manuscript.

General comments:

In my opinion the Abstract could be revised and modified. 

Response (2)

As suggested by the reviewer, we drastically changed the abstract.

The literature review seems to be very limited. I suggest authors to elaborate more on the topic and refer to many more scientific research related to the paper subject.

Response (3)

We thank the reviewer for the useful comment, identifying areas for improvement. As advised by the reviewer, we have extended the literature review in the Introduction.

The literature review should be based on the most recent scientific data. Now, more than 50% of cited references are old. Please improve that.

Response (4)

As suggested by the reviewer, new latest references have been added in the revised manuscript especially in the Introduction. In fact, the delineation of “new” and “old” literature is not so straightforward.  By now Chernobyl studies have 35 years of history. Sometimes one cannot but refer to data obtained in the late 1980s, 1990s, when major findings were made in the field of accidentally released radionuclide behavior in the sediment-water environment. Also, works such as Donigian et al., 1977; Knisel, 1980; Ahuja et al., 1981 and Whicker et al., 1990 are regarded as classical publications on the subject and are still valid and relevant.

It is not clear what is the hypothesis of performed study.

Response (5)

We thank the reviewer for the useful comment, identifying areas for improvement. Regarding the hypothesis, we proceed from the premise that the vertical profile in bottom sediments represents long-term dynamics of particulate concentration, given certain conditions are satisfied for sediment column formation (Lines 43-50).

The authors should enhance the discussion and highlight the importance of performed study in terms of the sustainability.

Response (6)

We thank the reviewer for the useful comment. As suggested by the reviewer, we revised the manuscript to highlight the importance of the performed study in terms of sustainability. The proposed method makes it possible to retrieve useful information which may not always be available otherwise, and in combination with the semi-empirical model, it can be used to predict contamination in the future. Further, the method can be improved and data can be updated, for example, with annual mean sedimentation rate parameterized through a sum of annual precipitation or other parameters.

The novelty and limitations of the performed research should be delineated.

Response (7)

As suggested by the reviewer, we revised the manuscript to clarify and emphasize the novelty and limitations of our research (Lines 49-50; 73-75; 88-91).

The title indicates that the study was be comprehensive, but after reading the text, the reader may have impression that is still lack of mentioned comprehensiveness, especially in regards to the literature review and the discussion on the outcomes.

Response (8)

We thank the reviewer for bringing up this issue.

We believe that the proposed approach is comprehensive in the sense that it can be applied to various water bodies and contaminants, enabling reconstruction of mid- and long-term dynamics of water body contamination. Having said that, the paper is focused on three specific water bodies in ChEZ and reconstructing the history of their contamination by Cs-137 over 32 years after the Chernobyl accident. Following the reviewer’s advice, we revised the title to read “Reconstruction of the long-term dynamics of particulate concentrations and solid-liquid distribution of radiocesium in three severely contaminated water bodies of the Chernobyl exclusion zone based on current depth distribution in bottom sediments”

Once again, we appreciate the reviewer’s careful and professional review.

Reviewer 3 Report

The topic of publication is very interesting.
However, I would like to write several additions and recommendations.
Lines 209-210 of the articles argue that deposits accumulate about even time evenly that, unfortunately, not so.
If we are talking about the mechanism of accumulation of precipitation, then, on the one hand, you should pay attention to the composition and size of accumulated precipitation, and on the other hand, climatic conditions in which this deposition accumulates. It is known that the amount of accumulated precipitation depends not only on the mode and temperature of surface collectors, but in the climatic conditions of the region's region - air temperature, direction and speed of wind movement, the amount of precipitation and their evaporation. It would be interesting to compare changes in climatic conditions with the results obtained.

Author Response

We thank the reviewer for the time spent on the review and careful reading of our text. We appreciate the insightful, constructive comments and suggestions which have been very helpful in improving the manuscript.

Please see below a combined point-by-point response to address all reviewers’ comments.

Reviewers’ comments are shown in blue Italic.

The topic of publication is very interesting.

Response (1)

We thank the reviewer for this positive feedback.

However, I would like to write several additions and recommendations.

Lines 209-210 of the articles argue that deposits accumulate about even time evenly that, unfortunately, not so.

If we are talking about the mechanism of accumulation of precipitation, then, on the one hand, you should pay attention to the composition and size of accumulated precipitation, and on the other hand, climatic conditions in which this deposition accumulates. It is known that the amount of accumulated precipitation depends not only on the mode and temperature of surface collectors, but in the climatic conditions of the region's region - air temperature, direction and speed of wind movement, the amount of precipitation and their evaporation. It would be interesting to compare changes in climatic conditions with the results obtained.

Response (2)

We thank the reviewer for bringing up this point. We completely agree with the reviewer’s statement that deposit accumulation rate depends on air temperature, direction and speed of the wind, precipitation, and water evaporation during the year, determining sedimentation seasonal and other short-term variability. Yet, we study long-term dynamics of sedimentation in the water bodies, using mean annual values, which smooth over variations in sedimentation rate within a specific year. This feature was demonstrated for a number of lakes across the world with a relatively non-uniform distribution of precipitation within a year (Kaminski et al., 1998; Nehyba et al., 2011; Putyrskaya and Klemt, 2007). We supposed that the same would be true for ChEZ water bodies under study and the obtained reconstruction results were not contrary to what we expected. Strictly speaking, the accuracy of the proposed reconstruction method can be improved in the future by accounting for annual precipitation variations from year to year. Parameterization of mean annual sedimentation rate through rainfall erosivity factor R, based on Universal Soil Loss Equation (USLE) (Wischmeier and Smith, 1978), can help to obtain even more realistic sedimentation history. The use of such a parameterization of sedimentation rate time dependence would reduce uncertainty in reconstruction results. At this stage, it is important for us to set out the concept underlying our method and in the future, it can be further developed and improved.

We added a corresponding text and relevant references on the subject in the revised version of the manuscript (Lines 319-325; 340-350).

Once again, we appreciate the reviewer’s careful and professional review.

Reviewer 4 Report

Measuring very low amounts of radiocesium-138 is rather easy  - even from the living human body. Cesium is chemically similar to potassium so that many organisms take cesium instead of potassium and cesium will be precipitated when microorganisms, plants, or animals and their residues will precipitate. Radioactive cesium has been found from the air after the nuclear weapon test up to some 1964 and some accidents in plants treating nuclear power fuel. The half-life time of cesium-137 is some 30 years so that a part of cesium-137 in our environments including the sediments contain this compound. As this paper presents a much higher amount of cesium-137 was liberated in late April 1986 from the Chernobyl accident and the river Pripyat and Chernobyl plant cooling pond got much of cesium-137 from this accident. Some extra cesium-137 has then arrived in ponds and lakes due to remediation processes as presented also in this paper. Since the whole drainage basin of Pripyat may have got high fallout of cesium and a part of this was taken by vegetation and then by other organisms it is clear that some of this cesium has later been precipitated to sediments of the studied lakes and the pond.

In line 57 you mention “sediments mixing negligibly”. Vertical mixing in small ponds or lakes may be low especially if there are no animals. The horizontal mixing may happen according to the flow rate of water. Especially if the flow rate is variable (rainy or dry seasons, ice melting) many factors affect where the sedimented matter will stay (Krylov, A.L., Nosov, A.V. & Kiselev, V.P. Analysis of accumulation factor of cesium 137 in bottom sediments of surface water bodies. Russ. Meteorol. Hydrol. 36, 33–39 (2011). https://doi-org.ezproxy.uef.fi:2443/10.3103/S1068373911010055). Therefore, cesium is not a very good indicator to study sedimentation rate in spite of the fact that there are such studies.

Could the horizontal mixing partly explain that the maximum of cesium-137 was in different depths? Anyhow, after the sedimentation has transported sediment the old sediment with cesium-137 shall no more be disturbed – since there might be also many other isotopes.

Increase highly the Fig 6! Set the subfigures one below others; so that you can increase them and the comparing is easier!  You could also increase figures 4 and 5.

Author Response

We thank the reviewer for the time spent on the review and careful reading of our text. We appreciate the insightful, constructive comments and suggestions which have been very helpful in improving the manuscript.

Please see below a combined point-by-point response to address all reviewers’ comments.

Reviewers’ comments are shown in blue Italic.

Measuring very low amounts of radiocesium-138 is rather easy - even from the living human body. Cesium is chemically similar to potassium so that many organisms take cesium instead of potassium and cesium will be precipitated when microorganisms, plants, or animals and their residues will precipitate. Radioactive cesium has been found from the air after the nuclear weapon test up to some 1964 and some accidents in plants treating nuclear power fuel. The half-life time of cesium-137 is some 30 years so that a part of cesium-137 in our environments including the sediments contain this compound. As this paper presents a much higher amount of cesium-137 was liberated in late April 1986 from the Chernobyl accident and the river Pripyat and Chernobyl plant cooling pond got much of cesium-137 from this accident. Some extra cesium-137 has then arrived in ponds and lakes due to remediation processes as presented also in this paper. Since the whole drainage basin of Pripyat may have got high fallout of cesium and a part of this was taken by vegetation and then by other organisms it is clear that some of this cesium has later been precipitated to sediments of the studied lakes and the pond.

Response (1)

We thank the reviewer for understanding the essence of our manuscript and for the positive feedback.

In line 57 you mention “sediments mixing negligibly”. Vertical mixing in small ponds or lakes may be low especially if there are no animals. The horizontal mixing may happen according to the flow rate of water. Especially if the flow rate is variable (rainy or dry seasons, ice melting) many factors affect where the sedimented matter will stay (Krylov, A.L., Nosov, A.V. & Kiselev, V.P. Analysis of accumulation factor of cesium 137 in bottom sediments of surface water bodies. Russ. Meteorol. Hydrol. 36, 33–39 (2011). https://doi-org.ezproxy.uef.fi:2443/10.3103/S1068373911010055). Therefore, cesium is not a very good indicator to study sedimentation rate in spite of the fact that there are such studies.

Could the horizontal mixing partly explain that the maximum of cesium-137 was in different depths? Anyhow, after the sedimentation has transported sediment the old sediment with cesium-137 shall no more be disturbed – since there might be also many other isotopes.

Response (2)

We thank the reviewer for bringing up this point. We agree that horizontal redistribution of sediments can take place along the current surface of bottom after their sedimentation. However, it is expected that such movement does not change the temporal representation of the Cs-137 vertical profile. As a result of such horizontal redistribution of sediments, sedimentation rates differ for different locations on the bottom. At the same time, the determination of sedimentation rate was not the purpose of our work. The aim of this work was to reconstruct particulate Cs-137 activity concentrations in water bodies on the basis of its appropriate vertical profiles in bottom sediments. Also, Glubokoe and Azbuchin lakes are closed lakes with weak water movements and flows. We collected sediment cores in the deepest locations of each water body where horizontal water flows are minimal.

We added correspondent discussion and provided reference in the revised version of the manuscript (Lines 143-144; 192-195).

Increase highly the Fig 6! Set the subfigures one below others; so that you can increase them and the comparing is easier!  You could also increase figures 4 and 5.

Response (3)

We appreciate the reviewer for this technical comment and recommendation. Figures 4-6 have been enlarged to the maximal possible size.

Once again, we appreciate the reviewer’s careful and professional review.
